# Lentiviral Vectors for T Cell Engineering: Clinical Applications, Bioprocessing and Future Perspectives

**DOI:** 10.3390/v13081528

**Published:** 2021-08-02

**Authors:** Roman P. Labbé, Sandrine Vessillier, Qasim A. Rafiq

**Affiliations:** 1Advanced Centre for Biochemical Engineering, Department of Biochemical Engineering, University College London, Gower Street, London WC1E 6BT, UK; q.rafiq@ucl.ac.uk; 2Division of Biotherapeutics, National Institute for Biological Standards and Control (NIBSC), Blanche Lane, Potters Bar, South Mimms EN6 3QG, UK; Sandrine.Vessillier@nibsc.org

**Keywords:** CAR, T cell, lentivirus, viral vector, bioprocessing, manufacture

## Abstract

Lentiviral vectors have played a critical role in the emergence of gene-modified cell therapies, specifically T cell therapies. Tisagenlecleucel (Kymriah), axicabtagene ciloleucel (Yescarta) and most recently brexucabtagene autoleucel (Tecartus) are examples of T cell therapies which are now commercially available for distribution after successfully obtaining EMA and FDA approval for the treatment of blood cancers. All three therapies rely on retroviral vectors to transduce the therapeutic chimeric antigen receptor (CAR) into T lymphocytes. Although these innovations represent promising new therapeutic avenues, major obstacles remain in making them readily available tools for medical care. This article reviews the biological principles as well as the bioprocessing of lentiviral (LV) vectors and adoptive T cell therapy. Clinical and engineering successes, shortcomings and future opportunities are also discussed. The development of Good Manufacturing Practice (GMP)-compliant instruments, technologies and protocols will play an essential role in the development of LV-engineered T cell therapies.

## 1. Introduction

Viruses are infectious agents composed of nucleic acids protected by a protein coat. These microbes cannot self-replicate, relying instead on the cells they infect to produce more copies of themselves by hijacking the host’s replication machinery [1]. Viruses are generally associated with disease, but genetic engineers have repurposed their biology to leverage their natural ability to modify or supplement the host cell genetic code, producing a new class of gene delivery tools known as viral vectors [2]. Gamma retroviral (GRV) and lentiviral (LV) vectors are derived from enveloped RNA viruses of the *retroviridae* family and have been widely used in research and in the clinic due to their ability to integrate genetic material to the host cell genome and stably express a transgene, making them particularly suited to transduce highly replicating cells such as immune cells [3]. Chimeric Antigen Receptor (CAR) T cell therapy is the most successful type of immunotherapy to date and the only T cell-based therapy to have reached the market. To produce CAR T cells, T lymphocytes are genetically modified to express a CAR receptor, conferring them the ability to identify and destroy cancerous cells [4]. CAR T cell therapy has shown promising results for the treatment of blood cancers, particularly CD19 positive B cell malignancies with three products, Tisagenlecleucel (Kymriah) [5,6], Axicabtagene ciloleucel (Yescarta) [7,8] and most recently Brexucabtagene Autoleucel (Tecartus) [9,10] obtaining market authorisation. The bioprocessing of both vector and cellular elements of these therapies are rarely considered simultaneously and are not necessarily produced by the same manufacturer. The biotechnology and pharmaceutical industry are increasingly involved in T cell engineered therapies and sponsoring clinical tests, attesting to the rapid development of this therapeutic space. In this review, we will present the fundamental biological principles and the bioprocessing of LV vectors for gene-modified T cell therapies, as well as recent clinical and engineering advances.

## 2. Lentiviral Vectors

### 2.1. Biological Principles

The two main classes of retroviral vectors are GRV and LV vectors. The main difference between both vectors is that GRV vectors can only infect dividing cells, whereas LV vectors can also infect resting cells [2,3,11].

LV vectors have been used in multiple pre-clinical applications for the treatment of hereditary diseases such as haemophilia B [12] and macular degeneration [13] by in vivo delivery. LV vectors have also been used for ex vivo transduction of cells for therapeutic applications, the most advanced clinical application being CAR T cell therapy [14]. This review will focus on discussing the use and applications of LV vectors for T cell therapy development and manufacturing.

LV vectors have been derived from the *retroviridae* family of viruses, most commonly the human immunodeficiency virus type 1 (HIV-1) and have been used for research and therapeutic applications. Variant LVs have also been developed using source viruses from different origins such as the simian immunodeficiency virus (SIV), the feline immunodeficiency virus (FIV) or the equine infectious anaemia virus (EIAV) [15]. LVs are an effective means of modifying eukaryotic cells by relying on the natural ability of retroviruses to integrate their genetic material to the host cell genome [16]. LVs can be modified to carry up to 10 kb genes and stably transduce a gene of interest, often referred to as the “expression cassette” [17]. The LV genome encodes three structural genes, fulfilling the same function in both wild-type virus and engineered vectors. The three genes are the group-specific-antigen (*gag*), polymerase (*pol*) and envelope (*env*) [18].

The *gag* gene codes for structural proteins of the virus including the matrix, capsid and nucleocapsid. These proteins are part of protein scaffolds acting as a protective layer for the viral RNA genome [19]. The *pol* sequence codes for enzymatic functions of the virus essential for its replication cycle. This includes protease, which cleaves HIV immature proteins into mature functional proteins; reverse transcriptase (RT), which reverse-transcribes the viral genomic single-stranded RNA (ssRNA) into double-stranded DNA (dsDNA); and integrase, which incises the host genomic DNA to integrate the viral dsDNA [20]. The *env* sequence codes for the envelope proteins which are responsible for HIV cell entry and tropism by interacting with the target receptors. The envelope protein complex is a heterodimer formed of the two proteins encoded by *env*, gp120 and gp41. Gp120 and gp41 interact with their target receptors CD4, CCR5 and CXCR4 through the domains protruding from the envelope membrane [21].

Additionally, the HIV genome codes for regulatory elements Tat and Rev and accessory proteins Nef, Vpr, Vif and Vpu. While regulatory elements are required for HIV replication, accessory proteins enhance viral replication and are associated with pathogenicity in vivo [22,23]. Finally, the HIV genome codes for essential sequences involved in functions such as nuclear export, packaging and genome expression. The Rev-Response Element (RRE) interacts with Rev to allow nuclear export of unspliced and singly spliced HIV RNA during viral replication [24]. The HIV DNA genome is flanked by two long terminal repeat (LTR) sequences generated during reverse transcription of the viral RNA and are essential for viral genome integration to the host cell DNA and viral genome expression [25].

LVs have been developed and based on the retroviral genome by combining its components into recombinant plasmid DNA (pDNA). The pDNA can then be transfected into producer cell lines. The LV genome has been modified to increase the genetic payload that can be packaged into the vector, inhibit replication, and limit pathogenicity while retaining function [26,27,28]. The first generation of LVs were the most similar to the wild-type viral genome. To limit the risk of creating replication competent lentiviruses (RCLs), the viral genes were loaded onto three separate plasmids. The system consisted of a packaging plasmid containing the structural *gag* gene and the regulatory proteins—an envelope plasmid containing the glycoproteins of gp160 or alternative viral envelopes to modify tropism (e.g., VSV-G)—and a transfer plasmid containing the cassette flanked by the two HIV LTRs. The packaging and *env* plasmids did not contain the LTR sequences and the ψ packaging sequence, replaced instead by a cytomegalovirus (CMV) promoter, to further prevent the formation of RCLs [29]. The second generation of LV production system removed the accessory proteins Nef, Vpr, Vif and Vpu from the packaging plasmid, as these are associated with disease progression, pathogenicity and propagation through human communities, but are not essential for viral functions such as reverse transcription, integration or maturation [30]. The third and most recent generation of LV production system modified the 5′ HIV LTR sequence of the transfer plasmid and replaced it with a strong viral promoter, such as the CMV or Rous sarcoma virus (RSV) promoters, allowing the removal of *tat* from the system and further preventing the formation of RCLs. Finally, the Rev element was removed from the packaging plasmid and loaded onto a new regulatory plasmid, resulting in a four-plasmid system for added safety against RCL formation [31]. All three generations of LV vectors and their corresponding plasmid systems have been summarised in Figure 1.

### 2.2. LV Bioprocessing

LV vector bioprocessing can be divided in two main phases referred to as upstream processing (USP) and downstream processing (DSP). USP of viral vectors refers to the steps during which viral particles are produced. This is achieved through the transfer of DNA into packaging cells which are used as vector factories [32]. During the process development phase, this is often done in the 2D cell culture and upscaled into bioreactor production for clinical and commercial production [33]. The cells used are immortalised cell lines, the most popular being the Human Embryonic Kidney cell line HEK293T [34]. HEK293T is an adherent cell line which involves challenges for scaling up, as most large-scale bioreactors are adapted to suspension cell culture [33,35]. Some alternatives to accommodate adherent cell culture, such as packed bed bioreactors, have been suggested as a solution for effective scale-out [36]. Alternatively, efforts in developing suspension variants of HEK293T cells have also been attempted to address this issue [37]. Viral vector bioprocessing begins with the expansion of the producer cell line, e.g., HEK293T cells. Cells are then transfected with plasmid DNA (pDNA) vectors to transfer the genes required for LV particle production. As they mature, LV vectors bud from the producer cell membrane and are released into the cell culture medium. Consequently, no cell lysis step is required to release the particles as for other non-enveloped vectors such as adeno-associated virus (AAV) vectors [38].

The DSP of the produced vector starts after the harvest of the producer cell culture supernatant. This product is highly heterogeneous and contains large amounts of process and product-related impurities. The clarification step is performed to remove large impurities such as aggregates and cell debris and can be achieved by filtration or centrifugation [39]. To help remove contaminating nucleic acids at later stages, this step may include a nuclease digestion [40]. The capture step aims at retaining and concentrating the LV particles and further removes process- and product-related impurities. Intermediate purification steps may be added to exchange the LV buffer, concentrate the product or remove specific impurities. This is followed by a polishing step to remove the remaining impurities, which are usually the hardest to isolate from the final product due to their similar physical properties to the LV vectors [41]. Both capture and polishing steps can be achieved through several combinations of downstream processing techniques which may include ultra-centrifugation, ultrafiltration/diafiltration (UF/DF), tangential flow filtration (TFF), liquid chromatography or sterile filtration [42]. The final step or fill-finish step consists of the final formulation of the LV vector, which may include a buffer exchange, sterile filtration, cryopreservation or excipient addition [43,44]. An example of a large-scale LV bioprocess is presented in Figure 2.

## 3. Gene-Modified T Cell Therapy

### 3.1. Biological Principles

One of the roles of T lymphocytes in immunity is to detect and destroy cells infected by harmful pathogens. During infection, T cells can detect foreign pathogens when presented with an immunogenic portion of a specific infectious agent called antigen [47,48,49]. This activation is mediated through the T cell Receptor (TCR), composed of the CD3 receptor and the CD4 or CD8 co-receptors. The CD4 co-receptor is used by helper T cells when interacting with antigen-presenting cells to activate their cytokine release function, which promotes the recruitment and activation of other immune cells, whereas the CD8 co-receptor is used by cytotoxic T cells to activate their cell-killing function. Other co-receptors, including OX40, CD28 and 4-1BB, enhance cell function such as proliferation, cytokine production and cell survival [50,51,52]. Once activated, T cells will start dividing rapidly, produce proinflammatory factors and activate their cytotoxic function to destroy infected cells. When the infection is resolved, the activated T cell population produced by clonal expansion will become exhausted, senescent and ultimately undergo apoptosis [53]. This process leaves only a small, long-lasting subset of memory cells capable of detecting the same antigen and undergoing rapid clonal expansion followed by differentiation to mount a strong and specific adaptive immune response [54,55].

During their maturation in the thymus, T cells that recognise self-peptides are removed in order to prevent autoimmune reactions [56]. Genetic and epigenetic modifications present in cancer cells lead to the deregulation of cell functions such as metabolism, proliferation, DNA repair, apoptosis, motility and attachment [57]. The immune system has the ability to detect some of these factors and use its killing function to eradicate cancer cells [58]. Modifications of the tumour microenvironment (TME) can also be a potential target for the immune system, as characteristics of the TME are clearly distinct from those of healthy surrounding tissue [59]. Lymphocytes which are attracted to the TME and infiltrate the tumour are referred to as tumour-infiltrating lymphocytes (TILs). Presence of these immune cells has been associated with better prognosis and chances of remission in cancer patients, and they have been used in clinic for autologous adoptive cell therapy [60]. However, cancer cells originate from damaged or mutated healthy cells and can therefore be difficult to detect for the immune system [61].

The Chimeric Antigen Receptor, or CAR, is an engineered TCR capable of binding to a target antigen. The CAR can be introduced into T lymphocytes to produce a CAR T cell with the ability to detect a specific antigen. The CAR is composed of the single chain variable Fragment (scFv) of the human IgG antibody. Three generations of CARs have been developed: the first generation was an scFv fragment conjugated to a CD3 signal to promote cytotoxic activity of the CAR T cell [62,63]. The second generation also harboured a CD28 signal to promote cell proliferation and cytokine production upon target binding [64]. The third generation of CAR added 4-1BB and OX40 regions to promote cell survival and prolong the CAR T cell’s in vivo persistence [65]. A fourth generation has recently been developed and contains specific cytokine signals which renders CAR T cells resistant to the immune dampening effects of the TME and further improves their in vivo lifespan [66,67].

### 3.2. T Cell Therapy Bioprocessing

Two main approaches have emerged from T cell adoptive therapy relying on two different source materials. The allogeneic approach relies on cells sourced from a “universal” source in a one-size-fits-all strategy. The autologous approach relies on cells sourced directly from the patient for a personalised medicine-type therapy. All commercially available T cell adoptive therapies have been relying on the autologous principle as the allogeneic approach has proven difficult to develop due to HLA typing and stem cell technology early advancement [68].

For most CAR T cell therapies commercially available or in clinical development, cells are isolated from the patient’s peripheral blood mononuclear cells (PBMCs) by apheresis or leukapheresis. This material contains a host of immune cells, including B cells, macrophages, monocytes, NK cells and T cells. The first manufacturing step is the isolation of the T cell subset from the PBMCs, which can be performed by magnetic bead selection or selective expansion [69,70]. Magnetic selection refers to the attachment of magnetic beads conjugated to antibodies which bind to a target T cell population which can be isolated when passed through a magnetic column. For CAR T cell development, anti-CD3, or a combination of anti-CD4 and anti-CD8, antibodies can be used to select the T lymphocytes [14]. After selection, cells are typically left to rest for several hours in culture medium. The cells can then be transduced using a viral vector containing the CAR cassette. This step allows the introduction of the CAR gene into the cell to produce fully functional CAR T cells [71]. An alternative to the viral transduction is transfection. T cells can be transfected with pDNA containing the CAR gene—by chemical transfection, using a transfection chemical agent such as polyethyleneimine (PEI)—or by electroporation, using an electric current to form pores in the cell membrane and allow the pDNA to enter the cell. These methods require large quantities of pDNA as they are carried out at the end of the expansion phase on much larger cell numbers and thus have had limited use in therapeutic applications [72].

When producing CAR T cells for therapeutic applications, gene delivery can be achieved by four main types of genetic material transfer: stable integration of the gene to the host T cell genomic DNA; non-integrating transient transfer, where the transgenic DNA persists as an episome; viral vector-mediated transduction; and non-viral transduction [4]. To reach the therapeutic dose of cells required to treat a patient, cells are kept in culture and allowed to expand. To stimulate T cell expansion, cells are activated using beads conjugated to monoclonal antibodies targeting CD3 and CD28 T cell receptors [73]. Post-activation cells can be cultured in bioreactors, which offer the possibility to expand cells in a closed system, at medium scale and with sampling and monitoring to control product quality throughout the process [74].

Development of CAR T cell therapy presents a few challenges including poor cell expansion, short in vivo lifespan, and significant side effects in treated patients [75]. Obtaining enough target T cells for delivering a functional therapy is one of the main challenges. Material isolated for preclinical drug development is often obtained from healthy donors, whereas material used in the clinic originates from individuals with advanced forms of cancer that would have undergone harsh therapeutic regimens including chemotherapy, radiotherapy, immunotherapy or a combination of all of these. Consequently, material for clinical use will be of lesser quality and higher variability. Finally, sampling of immune cells from patients in a critical state will have dramatic consequences on their health and chances of survival [69]. Therefore, effective cell expansion technologies are paramount to bringing CAR T cell therapy from “bench to bed” [14,76].

Despite much success, CAR T cell therapy has been shown to pose major safety challenges. This is due to the potential of the CAR T cell triggering an immune response cascade known as cytokine release syndrome (CRS) or “cytokine storm”. CRS is caused by the constitutive overactivation of T cells resulting in a massive release of pro-inflammatory factors, leading to the activation of more immune cells in a positive feedback loop. CRS symptoms include high fever, seizures, organ failure, low blood pressure, and systemic inflammatory response and can lead to death [75,77].

## 4. Applications of LV Vector Technologies for T Cell Engineering

### 4.1. Introduction

LV vectors are a well-established in vitro technology for research and development, and more recently they have been gaining popularity for clinical applications. Indeed, safety and efficacy of LV clinical applications has been proven for many clinical trials [78] with some products reaching market authorisation. The manufacturing process used for the production of autologous therapies commercially available is described in Figure 3. However, challenges such as CRS, neurologic toxicity, and overall cost need to be addressed to make adoptive T cell therapy a widely accessible therapeutic [75]. The development of Good Manufacturing Practice (GMP) compliant instruments, technologies and techniques are essential for the development of LV-engineered T cell therapies. In this section we will present some applications of LV vector technologies for T cell engineering. The following discusses currently-available CAR T cell therapies, technologies designed to integrate LV applications to T cell engineering, LV-based research and development tools for T cell therapy development, allogeneic approaches using LV modified donor T cells and alternatives to CAR T cell therapy.

Viral vector bioprocessing begins with the expansion of the producer cell line, e.g., HEK293T cells. Cells are then transfected with plasmid DNA (pDNA) vectors. Post transfection, the viral particles produced are harvested from the cell supernatant. The clarification step is performed to remove large impurities such as aggregates and cell debris; this step may include a nuclease digestion step. The capture step aims at retaining the LV particles and further removes impurities. The polishing step removes impurities which are usually with similar physical properties to the LV vectors. The fill-finish step consists of the final formulation of the LV vector.

Autologous CAR T cell bioprocessing starts with the isolation of peripheral blood mononuclear cells (PBMCs) from the patient by apheresis or leukapheresis. To isolate the T lymphocytes from the PBMCs, an enrichment step can be introduced using antibody-conjugated magnetic beads. T cells are activated using anti-CD3/CD28 antibody-conjugated beads. Post-activation, cells are transduced using lentiviral vectors, and the CAR therapeutic gene is integrated in the target T cell genome to produce fully functional CAR T cells. The activated T cells are then expanded in culture to reach the required cell number for the therapy infusion course. The cell material is then harvested and conditioned for cryopreservation at −120 °C. The material is then shipped back to the clinic to be administered to the patient through intravenous infusion.

### 4.2. Commercially Available Gene-Modified T Cell Therapies

Of the three CAR T cell therapies with market authorisations, brexucabtagene autoleucel (Tecartus, Kite Pharma Inc., Los Angeles, CA, USA) [9,10] and axicabtagene ciloleucel (Yescarta, Kite Pharma Inc.) [7,8] are transduced using GRV vectors, while tisagenlecleucel (Kymriah^®^, Novartis International AG, Basel Switzerland) [5,6] is transduced using a lentiviral vector. Brexucabtagene autoleucel and axicabtagene ciloleucel both use the same viral vector for the transduction of the transgene payload under the control of the MSCV promoter [7,9]. Tisagenlecleucel uses a unique LV vector and the transgene is under the control of the EF-1a promoter [5]. Although other gene transfer techniques for adoptive T cell therapy are currently under clinical assessment, none have yet gained market authorization, demonstrating the importance of retroviral vectors as gene transfer tools for cell therapy applications. All EMA and FDA approved therapies have been listed in Table 1.

The current protocols for autologous CAR T cell therapies have been established and are presented in Figure 3. This autologous approach represents a viable option for commercial products but requires further improvement to address global demand [14]. Due to cost and manufacturing capacity, LV-engineered T cell therapies currently commercially available are only considered as a second-, third- or fourth-line therapeutic option for patients that have failed to respond or relapsed following conventional chemotherapy, radiotherapy and immunotherapy treatments. Improvement of the bioprocesses and technologies required for the production of LV vectors and T cell engineering must both be considered in parallel to improve patient access. Recent advances have been proposed to address process bottlenecks, enhance functionality and improve safety of LV engineered T cell therapies.

Logistics involved in cell and gene therapy which include sourcing of patient material, production, shipping, conditioning, and storage of the drug product are seen as one of the main bottlenecks to making cell and gene therapy a viable therapeutic solution [79]. This 22-day bioprocess is currently being carried out in the United States, posing a major challenge for other markets including the European Union, where patients need to ship their cells across the Atlantic twice before receiving therapy, and outlining the logistics issues posed by the current approach to autologous T cell therapy. To alleviate this, Novartis is increasing its manufacturing capabilities in France and Switzerland [80].

The European Medicines Agency (EMA) PRIority MEdicines scheme (PRIME) and the United States Federal Food and Drug Administration (FDA) breakthrough therapy designation have been developed to facilitate market approval for medicines that demonstrate in pre-clinical studies a significant improvement on current standard of care [81]. All three therapies benefitted from this accelerated evaluation as these therapies have shown high efficacy for B cell malignancies and are addressing an unmet clinical demand [82,83].

Tisagenlecleucel received FDA and EMA approval in 2018 for the treatment of mantle cell lymphoma (MCL) [5,6]. T lymphocytes are modified using a lentiviral vector to transduce them with an anti-CD19 CAR, and modified cells are then infused back into the patient. Clinical investigation is still ongoing in Phase II (ELIANA, NCT02435849), with results available and scheduled to be completed in November 2022 [84,85]. Phase III (BELINDA, NCT03570892) is currently being performed but due to the successful accelerated approval status, tisagenlecleucel can be commercialised before its completion [86]. Encouraging results from the anti-CD19 CAR T cell approach have resulted in increased efforts to standardise protocols for experimental development and anticipate clinical requirements [18,87,88].

### 4.3. Limitations, Safety and Efficacy

Despite initial clinical successes of CAR T cell therapy in treating blood cancers, safety issues and improving efficacy in solid tumours are still obstacles that need to be addressed.

In addition to separating LV genes into separate plasmid DNA constructs to avoid RCL generation, several modifications to the LV vectors have been implemented to improve safety. One solution to mitigate the development of adverse effects after CAR T cell infusion is the inclusion of transgenes containing a “suicide gene” [89,90]. A “suicide gene” allows the selective killing of CAR T cells by the addition of a specific chemical agent that is otherwise non-toxic to unmodified cells. This technology has been progressed to the pre-clinical and clinical stage [91,92]. For instance, the anti-SLAMF7 therapy against multiple myeloma (MM) included a dimerization domain fused to a caspase-9 domain suicide gene to mitigate the risk of off-target activation, as SLAMF7 is expressed on healthy leukocytes, including NK cells. The authors have shown efficacy in vitro and in vivo, and the modified cells could be eliminated by the addition of the dimerizing agent AP1903 (rimiducid) [93].

Another associated risk to CAR T cell manufacturing is the transduction of other cell types with the transgene. The transduction of a single leukaemia B cell led to the development of a resistant transgenic clone in a patient treated with the anti-CD19 CAR T cell therapy tisagenlecleucel (Kymriah) [94]. This led to the masking of the target CD19 epitope due to the cis-integration of the CAR therapeutic gene during manufacturing. Although a rare event (1 patient out of 369 treated at the time), the use of integrating vectors with broad tropism can represent a serious safety concern. The vector used for this therapy was pseudotyped using a VSV-G envelope protein which confers a broad tropism to the vector [2]. The engineering of more specific chimeric envelope proteins has been suggested as a means to address this issue. Measles envelope-based chimeric protein is capable of targeting the CD3 receptor only present on T cells. In vivo data in a humanised mouse model showed the specificity of such vectors and even allowed in vivo gene delivery to produce functional CAR T cell transduction. Although more work would be required, this method could address potential off-target transduction during ex vivo transduction [95].

The development of resistant lymphoma B cells which do not express the CD19 antigen leading to the relapse of CAR T cell-treated patients has also been reported [96,97]. Transgene engineering has been proposed to address the development of resistant cancers. The introduction of two to three CAR constructs into each engineered cell has been tested in vitro and is under clinical investigation, with encouraging results on preventing the development of resistance to CAR therapy [98,99].

CAR expression level was shown to impact clinical outcome. The careful selection and design of promoters driving the expression of the CAR gene allows the improvement of CAR expression [100]. The different components of the CAR transgene such as the transmembrane and hinge domains has also shown to impact CAR expression. Efforts to optimise the design of intracellular and extracellular domains of the chimeric receptor have been under investigation [101].

Despite initial successes in treating blood cancers, CAR T cell therapy has encountered obstacles and limited success with treating solid tumours [102]. The physical properties of solid tumours as well as the tumour micro-environment (TME) restrict the ability of the CAR T cell to effectively access and kill the cancer cells [103]. The obstacles often cited are the difficulty for the cells to penetrate the tumour bed, the harsh physiological conditions within the tumour and the production of immunosuppressive modulatory proteins by the cancer cells. These barriers are not present in blood cancers as they do not form large tumour beds seen in solid tumour cancer types, thus explaining the success encountered with anti-CD19 CAR therapy.

Finally, alternatives to LV and RV vectors have been suggested to address the limitations and safety issues encountered. Electroporation is one such alternative which consists of the introduction of pDNA by passing an electric current through the cells [104]. The current temporarily generates pores in the cell as well as moving the negatively charge mRNA through the cell suspension and into the cell. Because of the non-integrating nature of this gene transfer technique, safety can be improved, as the expression of the transgene will slowly decrease as the mRNA is degraded and diluted through cell division. Although these techniques present safety advantages, long-term efficacy still needs to be demonstrated in human trials.

### 4.4. Development of LV-Based Tools for CAR T Cell Therapy Research

In addition to serving as effective and safe tools for CAR transgene transfer into T cells, LV vectors have been employed for developing in vivo and in vitro models. An essential part for developing in vitro and in vivo models is the development of target cells expressing the tumour-associated antigen (TAA) targeted by the CAR T cell therapy. Because patient material is a scarce resource and ethically cannot be used in every research and development study, immortalised cell lines have been a popular cancer model to test the CAR T cell performance in pre-clinical studies [105,106,107]. LV vectors have been utilised to rapidly modify immortalised cell lines to obtain stable expression of the target TAA and to use the modified cells for the development of in vitro killing assays, which are essential to demonstrate the therapy’s potency and specificity [99,106]. The same target cell lines can then be repurposed and engrafted to produce a tumour model in vivo in order to study the safety and efficacy of the therapy [99]. Murine models have played a major role for in vivo testing of CAR T cell therapies, and are used to test different aspects of therapeutic efficacy and safety [108]. Mouse models can be divided into four main classes: syngeneic immunocompetent, human xenograft, immunocompetent transgenic and humanised transgenic [109,110,111]. All these models are infused with human or mouse cancer cell lines to test CAR T cells and are essential to understanding tumour development and systemic immune responses to the therapy [108]. Providing evidence that the therapy developed in pre-clinical phases is both safe and efficacious is also a prerequisite to progressing to in-human clinical phases [112].

### 4.5. Development of Instruments Integrating LV to Current T Cell Bioprocesses

The autologous approach to CAR T cell therapy has proven to be effective; however, many challenges remain to be addressed. Bioreactor manufacturers have been designing platforms accommodating the requirements of GMP production in closed systems and including specific adaptations with T cell bioprocessing in mind e.g., the ambr 250 (Sartorius, Göttingen, Germany), Replicell™System (Aastrom Biosciences, Cambridge, MA, USA), Cocoon (Lonza, Basel, Switzerland) and Quantum Cell Expansion System (Terumo BCT, Lakewood, CO, USA) [113]. One of the most popular of these systems is the CliniMACS Prodigy bioreactor platform (Miltenyi, Bergisch Gladbach, Germany). It is a single-use, fully closed and GMP-compliant solution for T cell engineering. This bioreactor has a culture chamber which includes a centrifuge for medium exchange and formulation and can be operated in lower grade clean room environments. This platform includes viral vector application lines which allow the T cell therapy manufacturer to fully integrate LV transduction to the manufacturing process [114,115,116]. It is suited for an autologous approach, as the scale is only adapted for the production of a single dose per instrument. Although this solution does not allow scaled-up production, it represents a promising scale-out solution and could be used in a decentralised model where doses are produced closer to, or within, the clinical site [117]. This technology has been used on-site for Phase I clinical trials investigating a dual anti-CD19 anti-CD20 CAR T cell therapy for non-Hodgkin’s B-cell lymphomas. The 14-day manufacturing process fully integrated the transduction of the anti-CD19/CD20 CAR genes using an LV vector. Of the 8 patients treated, all therapeutic doses passed the release criteria, exceeded the cell number required for infusion and were successfully administered [118]. These technologies could provide a solution for improving product quality and production flexibility and could also reduce production costs, logistics complexity and product variability.

### 4.6. CAR T Cell Allogeneic Approach Using LV Vectors

Autologous CAR T cell therapy has proven to be an effective strategy for current therapeutic applications. The starting cellular material is sourced directly from the patient, mitigating risks associated with graft versus host disease (GvHD) [119]. Furthermore, the autologous nature of the material does not result in the production of anti-HLA donor-specific antibodies (DSAs) which can limit the number of doses and in vivo persistence of the whole organ transplant and allogeneic T cells [120,121]. Although autologous T cell therapies have proven to be easier to develop, major limitations render this approach difficult to scale up for reducing production time and cost. Starting material variability compounded with low cellular expansion potential renders a therapeutic dose difficult to achieve during the expansion of autologous T cells [122]. This is due to the nature of the material sourced from patients who have undergone multiple lines of treatment prior being eligible for cell therapy [123]. In addition, the autologous nature of the material means that each dose is specific to each patient, which rules out any option for large-scale production, leaving only bespoke scale-out options [69]. Due to all these limitations, increasing interest has been generated around the allogeneic approach in a one-size-fits-all model. In this case, the material would originate from a universal source such as genetically modified donor T cells, umbilical cord blood (UCB), placenta, immortalised cell lines, or induced pluripotent stem cell (iPSC)-derived T cells, opening the possibility to large scale production [124,125,126,127,128]. At present, only the genetically modified donor T cell and immortalised cell line approaches have progressed to the clinical stage, as the stem cell-derived approaches still require further pre-clinical optimisation [125,127].

The first proof-of-concept in-human testing of the donor-derived T cell allogeneic CAR T cell therapy was performed in 2015 in two paediatric patients at Great Ormond Street Hospital, London. The UCART19 technology was used in conjunction with lymphodepletion, including an anti-CD52 serotherapy, prior to the CAR T cell infusion. An LV vector was used to introduce the anti-CD19 CAR therapeutic gene, and a transcription activator-like effector nuclease (TALEN), a gene editing tool, was used to knock out the CD52 and TCR genes in the donor T cells to evade lymphodepletion and avoid GvHD. The results of this study showed that following allogeneic CAR T cell infusion, molecular remission could be achieved 28 days after infusion in both patients [125]. UCART19 was exclusively licensed by Cellectis to Servier, who sponsored two Phase I clinical trials started in 2016 and completed in 2020. In total, 7 children and 14 adults with B cell acute lymphoblastic leukaemia were enrolled. Published results from both trials showed that the most common adverse effects were CRS (91%) and neurotoxicity (38%) which resulted in two treatment-related deaths. Overall survival was 55% and progression-free survival 27% [129]. Building on UCART19′s initial success, Cellectis, Allogene and Servier begun investigating the use of TCR and CD52 KO CAR T cells in conjunction with an anti-CD52 recombinant antibody to treat relapsed or refractory large B cell lymphoma or follicular lymphoma. The administration of the allogeneic CAR T cells (ALLO-501) is performed after lymphodepletion of the patient’s own immune cells using fludarabine, cyclophosphamide, and a recombinant anti-CD52 antibody (ALLO-647). This technology is currently under investigation in a phase I, single-arm, open-label trial in 22 patients (NCT03939026) [130]. The preliminary data presented at the American Society of Clinical Oncology in 2020 showed ALLO-501 and ALLO-647 produced an objective response rate (ORR) in 12/19 patients (63%) with complete response (CR) in 37% of responding patients [131].

Although the allogeneic T cell donor approach has shown promising results, it still requires more process steps to knock out or counteract immunogenic T cell biological features that otherwise may result in GvHD. The integration of LV vector technology in the most advanced allogeneic CAR T cell therapy bioprocesses further demonstrate the relevance of this method for gene transfer in T cell engineering. If ongoing clinical trials result in market approval, demand on clinical grade LV vector and stress on global supply will further increase. This outlines the necessity for improved processes and technologies to increase the production and quality of LV vectors for T cell engineering.

### 4.7. Stable Production of LV Vectors to Address Increasing Demand in Vectors

In an endeavor to address global demand, efforts have been invested in improving current LV bioprocessing protocols and developing technologies to increase production. Large pharmaceutical companies have been investing in research and development of technologies designed to address bottlenecks. One of the main restrictions is the scale at which LV vectors can be produced using the HEK293T cell line. Although this HEK293T adherent cells transiently transfected with plasmid DNA LV systems can achieve high titres, the adherent nature of this cell line represents an obstacle to conventional scale-up solutions such as stir tank bioreactor production [132]. Efforts to address several of the bottlenecks presented by this method of production have been attempted in recent years. Stable production of LV vectors with dedicated packaging cell lines have been suggested as a potential solution to reduce cost associated with the use of large quantities of GMP-grade plasmid DNA required in transient production models [133]. In stable packaging cell lines (PCL), the viral genes required for LV production are stably transduced into the PCL genome. High titre-producing clones can be selected for vector production [33,134,135,136,137,138]. This approach has shown some encouraging results, though low titres compared to transient transfection still remains a major obstacle [139]. In addition to stable PCLs, the development of suspension variants of HEK 293T cells has also been suggested as an alternative [140]. Recently, Chen et al. (2020) combined both suspension and stable aspects in their cell line using a single stably-transfected construct producing comparable LV titres to transient transfection systems. This technology was tested in a stir tank bioreactor which offers potential for up-scaling production; CAR T cell therapy was cited as one of the potential applications [141]. Originally developed by the Telethon Institute for Gene Therapy (TIGET, Milan, Italy) and Cell Genesys (San Francisco, CA, USA), this technology was transferred to GlaxoSmithKline under the rare diseases strategic alliance and patented (patent number PCT/EP2016/078336), outlining the potential for future commercialisation [141,142].

### 4.8. Dedicated LV Technology for CAR T Cell Therapy Applications

In addition to increasing global production capabilities of LV vectors, development of dedicated LV particles for CAR T cell applications can improve therapy bioprocessing and biological function. When using LV vectors for gene delivery, the activation of T cells is essential to achieving high levels of transduction efficiency [143]. The activation of T cells also leverages the natural ability of immune cells to clonally expand when responding to an antigen. This biological feature of T lymphocytes is used to produce sufficient cell numbers required for treating the patient from the initial sample obtained by leukapheresis. This process is typically carried out using magnetic or polymer beads conjugated to anti-CD3 and anti-CD28 antibodies presented in Table 2 [69]. The LentiSTIM and RetroSTIM technology (patent number PCT/GB2016/050537) has been developed to combine both transduction and activation of CAR T cells. During the LV vector budding from the producer cell, some cellular proteins are carried along on the viral envelope [144]. This LentiSTIM particle is produced in cell lines expressing anti-CD3 and anti-CD28 membrane-bound mitogens, which are subsequently present on the LV envelope as it buds from the producer cell membrane. The same approach has also been employed to express biotin groups on the LV envelope to improve downstream purification [145]. This technology streamlines the CAR T cell production by reducing the reagents and process steps required and improving vector purity [146].

### 4.9. Alternative Approaches to CAR T Cell Therapy Using LV Technology

TCR T cell therapy is an alternative to CAR-transduced T cells for targeting malignancies. Instead of relying on a chimeric antigen receptor, this therapy relies on a modified T cell receptor [147]. The T cells are engineered to express a TCR with a specific affinity for a target antigen, and the manufacturing of this type of therapy is identical to CAR T cell therapy. The main advantage of TCR T cells is that the engineered receptor uses the same activation and immune pathways as a non-modified T cell, reducing the risk of CRS and neurotoxicity [148]. Adaptimmune Therapeutics, PLC is the proprietor of the Specific Peptide Enhanced Affinity Receptor (SPEAR) technology and is currently sponsoring or licensing technology in several clinical trials for the treatment of solid tumour cancers. The SPEAR T cells contain optimised TCRs and are manufactured using an LV vector encoding the TCR transgene [149]. The first completed Phase II trial was the “redirected auto T cells for advanced myeloma” trial sponsored by GlaxoSmithKline (NCT01352286), resulting in an ORR in 20/25 patients (80%) at day 42 and 11/25 patients (44%) after 1 year [149,150]. The SPEAR T cells are also under clinical testing in the Phase II SPEAREAD 1 (NCT04044768) and 2 (NCT04408898) trials sponsored by Adaptimmune, investigating the therapy ADP-A2M4 for the treatment of synovial sarcoma, myxoid liposarcoma and head and neck cancer [151]. T cells redirected for universal cytokine-mediated killing or TRUCKs are another type of alternative receptor to CARs. Sometimes referred to as the fourth generation of CAR T cells, TRUCKs have been developed to improve CAR T cell efficacy in solid tumours through cytokine-release. Upon binding of the CAR to its target antigen, specific cytokines are released from the gene-modified T cell, therefore counteracting immunosuppressant effects of the TME [152]. TRUCKs are currently under clinical investigation and can give an answer to improving CAR activity in solid tumours. For example, an anti-EGFR-IL12-CART is currently tested against metastatic colorectal cancer (NCT03542799) [153]. Specific LV technology has been developed to transduce all the elements required to produce a fully functional TRUCK using a single LV construct, further increasing the appeal of this class of T cell-based immunotherapy [154].

In addition to alternatives to chimeric antigen receptors, novel T cell subsets are considered as new potential therapeutic cells. Amongst new candidates, γδ T cells, natural killer T (NKT) and T regulatory (Treg) cells have been suggested as promising alternative cell types for adoptive T cell therapy and LV technology has been frequently employed for their engineering [155,156]. The γδ T cells account for 1–10% of the CD3+ T cell population. They are a highly conserved immune cell type and have shown attractive characteristics for adoptive T cell therapy applications. These include their potent graft versus tumour (GVT) activity as well as their low GVHD potential with allogeneic transplantation [157]. A study tracking the immune profile of 108 cancer patients up to 3 months after having received HSCT demonstrated that high levels of γδ T cells were associated with significantly higher overall survival and relapse-free survival [158]. Protocols and patents for γδ T cell culture and transduction using LV vectors have been published and developed to include adaptations for future manufacturing, such as serum-free medium culture [159,160,161]. Some studies have been progressed to Phase I testing, such as the IN8BIO Inc. technology NC200 clinical trial investigating the use of LV-modified γδ T cells to target glioblastoma multiform (GBM) in adults [162].

Invariant natural killer T cells or iNKT cells have also been suggested as a promising T cell subset for CAR T therapy. As γδ T cells, they are a rare subset of T cells and represent less than 1% of circulating PBMCs. They play an important immunomodulatory role and have shown to have strong anti-tumour activity [163]. A strong advantage of iNKT is the ability to transplant them without causing GVHD, making them a strong candidate for “off the shelf” applications. iNKT cells have been used in multiple clinical trials including therapy against solid tumours such as neuroblastoma. An anti-GD2 CAR construct has been transduced using a GRV vector into iNKT cells to target neuroblastoma. Preclinical data against showed in vivo efficacy with no significant toxicity, which led to progression to clinical phase [164,165]. The ongoing Phase I clinical study investigates the efficacy of CAR iNKTs called GINAKIT cells (NCT03294954) in combination with antibody therapy against relapsed or refractory high-risk neuroblastoma in patients between 1 and 21 years old [166]. iNKTs are also being tested against blood cancers, e.g., an anti-CD19 CAR-iNKT therapy for relapsed, refractory, high-risk B cell tumours is currently in phase I trials (NCT04814004) [167].

Regulatory T cells, or Tregs, are a subset of CD4-positive T cells, of which the principal role is controlling immune responses [168]. Rather than carrying out pro-inflammatory functions, Tregs have an inhibitory effect on other immune cell types, controlling immune responses to foreign pathogens and preventing autoimmune diseases. Tregs have been suggested for new promising therapeutic applications and expand adoptive T cell therapy beyond oncology [169]. Because of their natural ability to dampen immune responses, these cells are particularly suited for the treatment of autoimmune diseases and the prevention of GvHD in graft recipients [120,170]. The first in-human Treg clinical trial testing Sangamo CAR Treg product TX200-KT02 has been approved for Phase I/IIa (2019-001730-34). The Tregs are transduced using a LV vector to express a CAR receptor against the recipient’s HLA A*2 and prevent immune-mediated allograft rejection in patients with end-stage renal disease (ESRD) [171].

## 5. Conclusions

LV vectors have been instrumental in the development of gene modified T cell therapies. These retroviral vectors have represented a safe and effective tool for gene transfer and have been applied to many of the development steps required for the recent progression of CAR T cell therapies to the market. Not only have LV vectors been directly applied for the transfer of the CAR therapeutic gene, they have also been employed to develop target cells required for in vitro and in vivo testing. The importance of LV vectors in T cell therapy development has been reflected by not only their integration into T cell therapy protocols but also instrument design used in their manufacturing. The advancement of therapies through later phase clinical trial and access to fast-track schemes set up by regulators, as well as active involvement from the biotechnology and pharmaceutical industry, foretells the future importance of T cell therapies in the oncology therapeutic arsenal. Although alternatives for gene transfer have been proposed, LV and GRV vectors are currently the most advanced methods for gene-modified T cell therapy manufacturing.

## Figures and Tables

**Figure 1 viruses-13-01528-f001:**
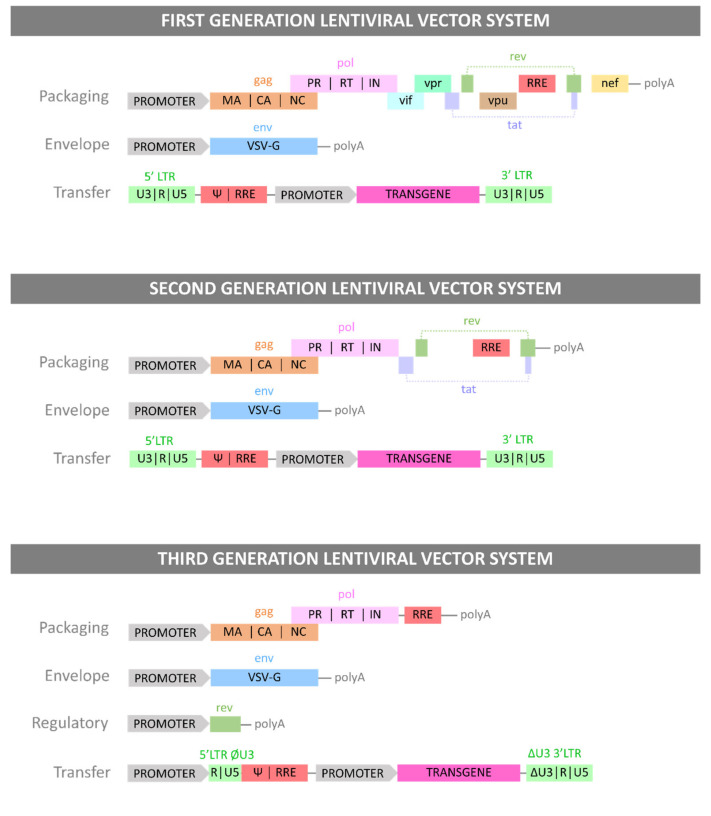
The three generations of LV plasmid systems. Each Generation of LV vector is presented with the plasmid constructs necessary for its production and with the genes each plasmid carries.

**Figure 2 viruses-13-01528-f002:**
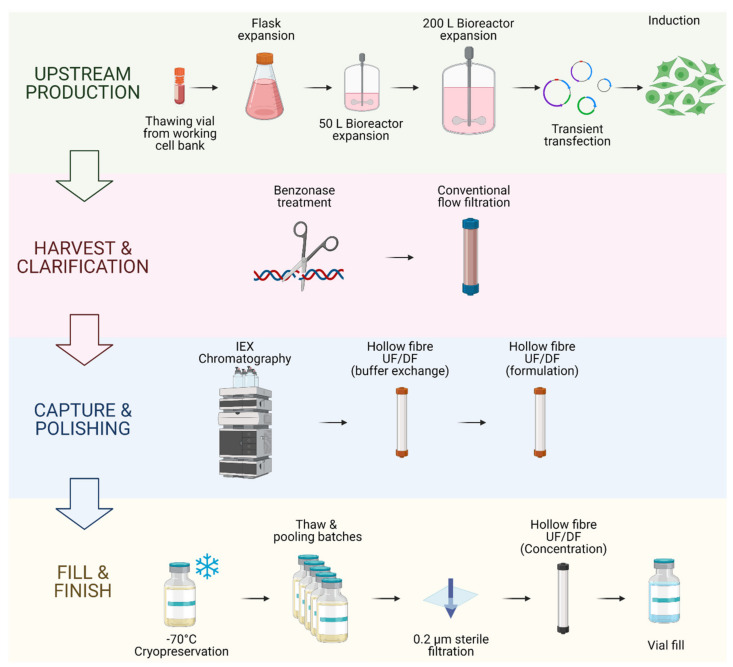
Example of an end-to-end upstream and downstream bioprocess for GMP-grade LV vector production. This figure describes the bioprocess patented by Oxford Biomedica (Oxford, UK) for the production of their GMP-grade lentiviral vector using their suspension-adapted, serum-free HEK293T producer cell line. This process includes an inducible plasmid system dependent on sodium butyrate. Each batch is individually cryopreserved and stored until enough material is produced, to then be combined, filtered and concentrated for final formulation [14,45,46].

**Figure 3 viruses-13-01528-f003:**
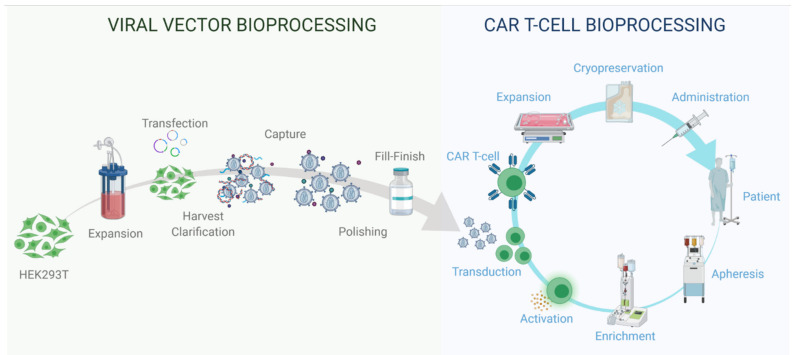
Viral vector and autologous CAR T cell therapy bioprocessing.

**Table 1 viruses-13-01528-t001:** EMA and FDA approved gene-modified T cell therapies (accessed on 4 January 2021).

INN ^1^ (Commercial Name)	Manufacturer	Application(s)	Therapy Type	Market Approval		Price per Dose (USD ^4^)		Reference
				EMA ^2^	FDA ^3^			
brexucabtagene autoleucel (Tecartus^®^)	Kite Pharma Inc. (Gilead)	Mantle cell lymphoma	CAR T cellGRV vector	2020	2020	$	373,000	[9,10]
tisagenlecleucel (Kymriah^®^)	Novartis AG	Acute B-cell lymphoblastic leukaemia	CAR T cellLV vector	2018	2017	$	475,000	[5,6]
axicabtagene ciloleucel (Yescarta^®^)	Kite Pharma Inc. (Gilead)	B cell lymphoma	CAR T cellGRV vector	2018	2017	$	373,000	[7,8]

^1^ International Non-proprietary Name—^2^ European Medicines Agency—^3^ Food and Drug Administration—^4^ United States Dollar.

**Table 2 viruses-13-01528-t002:** GMP compliant T cell activation technologies.

Product	Activation Method	Antibody Scaffold
Dynabeads (Gibco)	CD3/CD28 Antibody-mediated	Magnetic beads
TransAct (Miltenyi Biotec)	CD3/CD28 Antibody-mediated	Polymer beads
Cloudz (Bio-Techne)	CD3/CD28 Antibody-mediated	Dissolvable beads

## Data Availability

This review article did not include original data, all material cited has been referenced in text and in the bibliography.

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
