# Peer review of "Lentiviral Vectors for T Cell Engineering: Clinical Applications, Bioprocessing and Future Perspectives"

_viruses, 2021, doi:10.3390/v13081528_

Round 1

Reviewer 1 Report

This manuscript entitled “Use of lentiviral vectors for T-cell engineering” provides a review of the applications of retro/lentiviral vectors for chimeric antigen receptor (CAR) gene modified T cell therapies. The overall strength of this review manuscript is that it provides comprehensive information for the application of retro/lentiviral vectors to CAR T cell therapies, including vector biology, bioprocessing, clinical applications of the vectors and success of anti CD19 CAR T cell therapies for a handful of leukemic diseases. It provides the current preclinical research developments and the current challenges

It would be more informative if the reviews could provide more in-depth information for the followings.

1) It would be more informative to show the vector maps of different generations of CAR expressing lentiviral vectors. What is the best current configuration, promoter choice for the optimum CAR expression, how to express multiple genes including a suicide gene, 2A peptides vs multiple promotors? Does higher CAR expression corelates with more robust CAR T cell cytotoxic effects, clinical outcome?

2) It would be very informative to describe why the CAR T cell therapy is successful to only a few leukemic diseases, for example anti CD19 CAR, but not other majority of diseases? What are the limitations of lenti/retro vectors (for examples, low CAR gene expression, low vector titer?) and how to overcome these limitations to other cancers including solid cancers and infectious diseases (HIV and others).

3) The safety feature of retro/lenti vectors are missing. For examples, several suicide genes have been incorporated into the vectors to eliminate CAR T cells in case of sever adverse effects.

4)The description of new candidates, for example γδ T-cells and Treg cells are very informative and in depth, however, the information for Natural Killer T (NKT) (and iNKT cell) mediated CAR therapy are missing. Inclusion of these information would increase the significance of this review manuscript.

Author Response

Dear Reviewer,

We would like to thank you for the time and effort in reviewing the manuscript and for the amendments suggested. Please find below our point-by-point response to the points raised as well as an updated manuscript with additions/amendments in track changes

Comment number one has been addressed and a dedicated figure replaced the table previously presented. The purpose of the figure is to recapitulate the main components of an LV to aid the understanding of our article in a concise manner. The review references the publications presenting the detailed map of the different generations of LV vectors within the text.

Comment number two has been addressed by the addition of section 4.3 entitled “Limitations, safety and efficacy”. In this section, we discuss the limitations of lentiviral vector technology, safety considerations and alternatives. The promoters driving the CAR transgene expression currently in use for clinical applications have been added to section 4.2.

Comment number three has also been addressed in the newly added section 4.3, where we discuss and illustrate with clinical and pre-clinical examples the use of suicide genes to improve the safety of CAR T cell therapy.

To address comment number four, we added NKT and iNKT applications to section 4.9. In addition, a paragraph on TRUCKS has also been incorporated in this same section.

Reviewer 2 Report

In this review, the authors reviewed the biological principles as well as the bioprocessing of Lentiviral (LV) vectors and adoptive T-cell therapy. Overall the manuscript was written in a clear and concise manner.   

Main concerns:

  1. In addition to the advantages, the disadvantages of LV vectors need to be discussed.
  2. An overall comparison of current vectors using for adoptive T-cell therapy should be presented.
  3. The clinical uses of LV vectors may be listed and discussed. This will be very attractive for the readers.

Author Response

Dear Reviewer,

We would like to thank you for the time and effort in reviewing the manuscript and for the amendments suggested. Please find below our point-by-point response to the points raised as well as an updated manuscript with additions/amendments in track changes

Comment number one has been addressed by the addition of section 4.3 entitled “Limitations, safety and efficacy”. In this section, we discuss the limitations of lentiviral vector technology, safety considerations and alternatives.

Current vectors used for adoptive cell therapy are exclusively retroviral vectors (LV and gamma RV). Comment number two would be difficult to address as the vector technology used in commercially available therapies is proprietary and often not publicly disclosed. The promoters driving the CAR transgene expression currently in use for clinical applications have been added to section 4.2.

The scope of this review is to recapitulate the use of lentiviral vectors in the context of ex vivo T cell engineering and therefore did not cover other LV vector applications such as in vivo gene therapy. We do agree that understanding other uses of LV technology could aid in the comprehension of the review and have therefore added a short overview of alternative uses for LV vectors along with a few examples in sections 2.1 and 4.3.

Round 2

Reviewer 2 Report

The authors well addressed my previous comments.